# Chrysippus and the First Known Description of Cataract Surgery

**DOI:** 10.3390/medicines7060034

**Published:** 2020-06-22

**Authors:** Juliusz Grzybowski, Andrzej Grzybowski

**Affiliations:** 1Faculty of Fine Arts and Pedagogy, Adam Mickiewicz University Poznań,Nowy Świat 28-30 Street, 62-800 Kalisz, Poland; juliuszprotazy@wp.pl; 2Department of Ophthalmology, University of Warmia and Mazury, 10-561 Olsztyn, Poland; 3Institute for Research in Ophthalmology, Foundation for Ophthalmology Development, 60-554 Poznan, Poland

**Keywords:** history of ophthalmology, cataract surgery, Chrysippus

## Abstract

Although the origin of cataract surgery is unknown, the earliest identified mention of cataract surgery comes from Chrysippus in the 3rd century B.C.E. This historical review analyses this first description of cataract surgery from both philosophical and linguistic perspectives, within the original context in which early cataract surgeries were performed, as well as within the context of contemporary medical knowledge.

## 1. Introduction

The place and time in which cataract surgery originated are uncertain. It is often stated that the earliest description of cataract surgery comes from Sushruta, who probably lived in the 6th century B.C.E. However, the earliest surviving manuscripts of his work, “Sushruta Samhita: Uttara Tantra” date from the Common Era [1].

Thus, the earliest available description of cataract surgery comes from Chrysippus and was written in the 3rd century B.C.E. Interestingly, Chrysippus was known as a philosopher rather than as a physician. The aim of this study is to analyse the first mention of cataract surgery from both philosophical and linguistic perspectives within the original context in which early cataract surgeries were performed, as well as within the context of contemporary medical knowledge.

## 2. Methods

In his commentary on Aristotle’s “Categories”, dating from the 6th century C.E., Simplicius of Cilicia included a description of cataract surgery written by Chrysippus, a Greek Stoic philosopher who lived in the 3rd century B.C.E. (Figure 1) Both the fragment included in the text by Simplicius and the incompletely preserved text by Chrysippus examine the categories of privation and possession (gr. Steresis, στέρησις; and heksis, ἕξις). Cataract surgery was not the main area of interest for Simplicius or Chrysippus, and was used merely to further the philosophical discussion. Thus, the first written record of cataract surgery in the history of European literature is not a medical text but a philosophical one. This fact may raise doubts as to the medical competency and credibility of Chrysippus. The main purpose of this article is to thoroughly examine the fragment of the text by Simplicius in which the author cited Chrysippus. The analysis will focus on the linguistic aspects of the text within the context of philosophical studies conducted by Simplicius. 

The works of Aristotle are relevant for several reasons. First, Simplicius comments on the work of Aristotle and it is Aristotle, in fact, who is the main object of his study, not Chrysippus. That is why it is necessary to establish whether Chrysippus is treated by Simplicius with due diligence or plays merely an auxiliary role in his text. Second, the works of Aristotle functioned as a point of reference in natural sciences and also with regard to terminology both in classical antiquity as well as throughout the Middle Ages. The categories used by Aristotle, who preceded Chrysippus by a century, will be of use in establishing a timeframe not only for the text written by Chrysippus but also, indirectly, for establishing when the Greeks started to perform cataract surgeries. 

Based on the current knowledge of medical procedures available by the time of Chrysippus, an attempt will be made to propose the most reasonable hypothesis with regard to Chrysippus’s source of information about the procedure. In short, the article will aim to establish whence Chrysippus learned about cataract surgery.

## 3. Results

Below is the fragment by Simplicius:

“Aristotle] solves such disputes too by adding another difference between them, namely that contraries [can] change round in the things capable of receiving them, but it is not possible ‘that a change’ from possession or privation ‘to the other one come about’ (13a19). For although blindness comes about from sight, [change] does not[occur] in the reverse direction as well. And because of this Chrysippus raised the question whether those suffering from a cataract (hypochythentas,ὑποχυθέντας) but able to recover sight after a couching of the eye (ekparakentheseos,ἐκπαρακεντήσεως) should be called blind, and [he raised the same question] in the case of those whose eyelids are [naturally] shut: for since the capacity [to see] exists, they resemble someone [voluntarily] keeping his eyes shut, or someone prevented by a screen (parapetasma,παραπέτασμα) from seeing, since if this [screen] is removed (aphairethentos,ἀφαιρεθέντος) he is in no way prevented from seeing. So it is not from privation to possession that such a change comes about. But [Aristotle] is here considering the kind of privation which consists in a disability. For from such a [privation] there is no return to the [corresponding] possession [2,3].”

### 3.1. Linguistic Analysis

The word that is used by Simplicius to describe the cataract is derived from the verb hypocheo(ὑποχέω). In its literal sense, it means “to pour under” or “to spill”. In its passive form, it means “to be spilled” or “to be spread beneath something”. It is in that sense that the author of “On the Cosmos”, by some attributed to Aristotle, used the word: “Next under this is spread the air” (ἑξῆς δὲ ταύτης ὁ ἀὴρ ὑποκέχυται) [4]. In the surviving literature, the first instance of the word, in its medical sense, can be found in the fragment by Chrysippus cited by Simplicius. 

The word that Chrysippus used to denote the cataract surgery, parakentesis (παρακέντησις), is a noun derived from the verb parakenteo (παρακεντέω) that means, according to Lidell and Scott’s “Greek-English Lexicon” [5], “to pierce or poke at the side”, close to the present meaning of paracentesis.The oldest use of the word in a non-medical sense can be found in a text by Theophrastus (370-287 B.C.E.) in which he wrote about charcoal burning: “They cut and require for the charcoal-heap straight smooth 1 billets: for they must be laid as close as possible for the smouldering process. When they have covered the kiln, they kindle the heap by degrees, stirring it with poles (parakentountes obeliskois, παρακεντοῦντες ὀβελίσκοις). Such is the wood required for the charcoal-heap” [6]. The principal part of the verb parakenteo (παρακεντέω) is kenteo (κεντέω)—to prick or stab. Thus, it can be concluded that the cataract surgery required some form of stabbing. The precise location of such stabbing is not known, although the prefix para (πάρα)—at or to one side of/beside—gives us a clue. However, it is hard to tell whether the prefix refers to the location of the stabbing or, as in the fragment by Theophrastus, to the act of stabbing side by side. 

Simplicius gives us another clue. Although we cannot be sure to what extent Simplicius uses his own words and not the words of Aristotle or Chrysippus, he offers a commentary in which he compares a cataract to a curtain (parapetasma, παραπέτασμα) which has to be removed or torn away (aphaireo, ἀφαιρέω). 

### 3.2. Simplicius and Chrysippus

Simplicius was born in Cicilia c. 490 C.E. and died c. 560 C.E., although the date of his death cannot be known for sure. In Alexandria, he was a disciple of Ammonius Hermiae. Later, upon moving to Athens, he became a disciple of Damascius, the last scholar of the School of Athens. In 529 C.E., the emperor Justinian closed the School of Athens, forbidding pagan philosophy. Damascius, together with Simplicius and other disciples, left Athens and resolved to seek protection at the court of the Persian king Chosroes [7]. Some scholars believe that having spent some time in Persia, Damascius returned to Athens. Others believe that after leaving Persia he moved to Harrana, Syria. Nevertheless, it was at that time that Simplicius wrote his commentaries to Aristotle’s “On the Heavens”;“Physics”; and, most importantly, “Categories” [8,9,10].

Chrysippus (c. 277–208 b.c.) (Figure 2), Athenian philosopher and, according to Diogenes Laertius, “the son of Apollonius”, came either from Soli or from Tarsus, as Alexander relates in his “Successions”. He was a pupil of Cleanthes. Before this, he used to practise as a long-distance runner; afterwards, he came to hear Zeno, or, as Diocles and most people say, Cleanthes.Then, while Cleanthes was still living, he withdrew from his school and attained exceptional eminence as a philosopher [11]. Chrysippus specialised in logic and, as was typical of Stoics, ethics. For both Chrysippus and the Stoics in general, the source of happiness was moral beauty achieved through divesting oneself of any desires. This indifference of sorts was called apathy (apatheia) by the Stoics. We know that Chrysippus wrote about the theory of vision in “Physics”, although the treatise was lost, and its content is known only by indirect evidence. According to Diogenes Laertius, “They [Stoics] hold that we see when the light between the visual organ and the object stretches in the form of a cone: so Chrysippus in the second book of his Physics and Apollodorus. The apex of the cone in the air is at the eye, the base at the object seen. Thus the thing seen is reported to us by the medium of the air stretching out towards it, as if by a stick.” [11].

Another prominent figure of the period named Chrysippus was Chrysippus of Cnidus (4th–3rd c. B.C.E), a physician and the son of Aristagoras. This Chrysippus taught many physicians, among others Erasistratos and Aristogenes. It is believed that his writings and contributions were later described as belonging to his pupils; many of Erasistratos’ contributions are believed to belong originally to him. The situation is even more complicated because his grandfather was named Chrysippus and was also a physician. Moreover, the son of Chrysippus of Cnidus was named Chrysippus, and one of Erasistratos pupil’s sons was also named Chrysippus.It shows some problems related with precise recognition of the person. It is argued that Chrysippus of Cnidus was not in favour of phlebotomy, advocated starving, and studied the pulse. [12]. However, his writings were already very rare in the times of Galen.Diogenes Laertios [11] who lived a century later could deliver only one of his titles: “Treatments for Sight (τὰθεραπεύματα [...] ὁρατικά)”. There is no other information available on his works related to vision [12].

## 4. Discussion

To which Chrysippus does Simplicius refer: Chrysippus of Soli or Chrysippus of Cnidos, a Greek physician from the Alexandrian school? All of the available evidence points in the direction of Chrysippus of Soli. The fragment cited by Simplicius pertains to logical considerations. What is more, it is not the only fragment by Chrysippus cited by Simplicius in his commentary on Aristotle’s “Categories”. Simplicius uses cataracts merely as an example, an illustration of the problem. Cataracts are not the object of his study. What is more, it is highly probable that Simplicius used the original work of Chrysippus, not just an excerpt or a compilation thereof. 

Simplicius cites Chrysippus throughout several pages of his text (98 Γ–102 Ζ). Therefore, it seems highly probable that he had the source text right before him. It is not the only quotation from Chrysippus in the “Commentary to Aristotle’s Categories”. What is more, fragment 178 contains a passage where Chrysippus is, alongside Aristotle, cited by Simplicius as the main source.

Although Stoics hold a prominent position in the history of philosophy, their presence in the history of medicine is rather accidental and usually takes the form of a medical metaphor used for the purpose of philosophical ruminations [13], as in the case of Chrysippus. We owe them the development of logics and the concept of cosmopolitan politics, supported by Chrysippus, that were appreciated not earlier than in the 20th century. 

Stoics, Chrysippus among them, were interested in medicine only to the extent that it pertained to themselves personally. Stoics were usually interested in those subjects that were at least in some way connected with the philosophical questions they pursued, be it within the field of logic, ethics, or physics. As a result, it was not possible for the Stoics to avoid, in their philosophical pursuits, the question of the anatomic location of the soul [13,14,15,16,17,18]. We know that Chrysippus knew Praksagoras and, like Praksagoras, located the soul in the heart. However, it seems that Chrysippus was interested in medicine only as a means to an end. Interestingly, Chrysippus admitted that he “lacked experience in the questions of anatomy (ἀπείρως ἔχειν τῶν ἀνατομῶν)” [15,18,19]. When it comes to the predecessors of Chrysippus in the Stoic school, the catalogue of texts by Zeno of Elea does contain, unfortunately unpreserved, the treatise “Of Vision” (Περὶ ὄψεως), but the author was probably interested in a broadly defined theory of vision—i.e., the mechanisms of creating the objects of vision, rather than the physiological and medical aspects thereof. Although it is highly unlikely that Zeno wrote about questions of a medical nature, that possibility cannot be excluded. 

Chrysippus was an extremely prolific writer. According to Diogenes Laertius, “In industry he surpassed every one, as the list of his writings shows; for there are more than 705 of them” [11]. Such industriousness had a negative impact on the philosopher’s style. According to Diogenes Laertius, Chrysippus wrote down everything that came to his mind, often repeating himself and quoting other authors in such a way and to such an extent that it was difficult to establish whether his texts were written by him or by someone else [11]. What we also learn from Laërtius is that not only would Chrysippus write stories about the conjugal life of Zeus and Hera that were too obscene to quote, but he would also prove theses that were decadent, to say the least: “in his Republic he permits marriage with mothers and daughters and sons. He says the same in his work On Things for their own Sake not Desirable, right at the outset. In the third book of his treatise On Justice, at about line 1000, he permits eating of the corpses of the dead” [11].

The term hypochyma (ὑποχῦμα), denoting a cataract, which would later be found in the works of Dioscorides and Galen among others, as well as the term hypochysis (ὑποχῦσις), also to be found in Dioscorides and Aelianus, are both derived from the verb hypocheo (ὑποχέω) that Chrysippus used when he was writing about the cataract. Interestingly, neither hypochyma, nor hypochysis can be found in the works of Aristotle, even in the long fragment from the treatise “On the Generation of Animals” [20], in which Aristotle wrote about the eyes. Verses 780 a 14-25, in which Aristotle talks about eye diseases, including glaukoma (τὸγλαύκωμα) and nyktalops (ἡ νυκτάλωψ), are especially important. It is possible that Aristotle used the word glaukoma in the meaning of “cataract” [21]. However, one cannot forget that the Latin term for cataract, suffusio, follows the structure of the Greek term hypochyma, not glaukoma (hypo – sub → suf; hyma –fusio). In the available translations of the works of Aristotle, glaukoma (γλαύκωμα) is translated either as the cataract or as glaucoma. The question requires further analysis. However, Aristotle did write about age-related eye diseases and connected glaukoma (γλαύκωμα) with old age. Nyktalops (νυκτάλωψ), on the other hand, was, in the opinion of Aristotle, a disease observed among rather young people. Therefore, it seems that even if the word glaukoma (γλαύκωμα) refers to glaucoma and not the cataract, the text by Aristotle does contain a fragment in which the term hypochyma (ὑποχῦμα) could or even should be used. What is more, Aristotle did not use the verb hypocheo in any sense connected with eye diseases. Aristotle was probably not aware of that meaning of the verb and perhaps that is why Simplicius, in his commentary on Aristotle’s works, used the example from the text by Chrysippus instead.

Apart from the fact that the word hypochyma was not used in the treatise “On the Generation of Animals”, we have one more argument supporting our thesis. In the end, Simplicius comments on Aristotle’s “Categories”, a text that is preserved in its entirety. Let us hear what Aristotle has to say about this: “Privation and possession are spoken of in connexion with the same thing, for example sight and blindness in connexion with the eye. To generalise, each of them is spoken of in connexion with whatever the possession naturally occurs in. We say that anything capable of receiving a possession is deprived of it when it is entirely absent from that which naturally has it, and absent at the time when it is natural for it to have it. For it is not what has not teeth that we call toothless, or what has not sight blind, but what has not got them at the time when it is natural for it to have them. For some things from birth have neither sight nor teeth yet are not called toothless or blind” [22]. Although we cannot analyse the ruminations of Aristotle any further, it is worth adding that Aristotle uses vision loss as a perfect illustration of privation (steresis, στέρησις). Thus, Simplicius, by citing Chrysippus, agrees with the approach proposed by Aristotle. Aristotle juxtaposes privation with possession (heksis, ἕξις). Possession, in his view, is inseparable from a disposition to perform an action. Within the context of ethics, Aristotle uses the category of heksis (ἕξις) to describe a state which makes a man perform ethically specific acts—e.g., brave or generous acts. A man with such a permanent disposition not only may but must, in fact, behave in a certain way, as the permanent disposition becomes his nature. Thus, if a cataract could lead to the loss of vision, it would render Aristotle’s argument incomplete and his example imperfect. “With privation and possession, on the other hand, it is impossible for change into one another to occur. For change occurs from possession to privation but from privation to possession is impossible; one who has gone blind does not recover sight nor does a bald man regain his hair nor does a toothless man grow new ones” [22]. We may safely assume that if only Aristotle had heard about such a type of vision loss that could be reversed, he would surely use that as an example in his text. Aristotle, unlike Chrysippus, studied medicine and became a philosopher as an educated physician, as is suggested by the 10th book of Aristotle’s “Zoology”, which is a medical work on infertility. That is why Simplicius, looking for an example illustrating the relationship between privation and possession, uses the works of Chrysippus and not Aristotle. 

### 4.1. Chrysippus’s Credibility

Chrysippus uses terminology that can also be found in medical treatises from later periods. That is why it is highly unlikely that his text is one of the “fantastic” stories so common in the ancient world (e.g., Aristotle’s “Mirabilia”). In other words, if Chrysippus confabulated, he wouldn’t have used a term that was later used by specialists in the field. Chrysippus himself was not a specialist. First of all, it is hard to imagine Chrysippus, apparently uninterested in medicine, to be the first citizen of Athens to hear about cataract surgery. Secondly, it is highly unlikely that he would come up with a name for the disease, taking into consideration that contemporary translators of Aristotle often use the terms cataract and glaucoma interchangeably, and that the name coined by the philosopher would be commonly adopted by the medical community. It was probably some other Greek who was first to use the verb hypocheo in reference to the cataract and yet another who used an equally specialist term of parakentesis, which can be found in the works by Galen. What is more, Chrysippus’s argument would make sense only if cataract surgery had been a commonly known medical procedure. A reader must have known that it is a type of vision loss that can be reversed. That is why someone suffering from a cataract cannot be called a blind man because, according to Chrysippus, a blind man is someone who has lost his vision irreversibly. Chrysippus’s argument would have been completely without substance if cataract surgeries had not been performed at all or with at least some degree of success. Taking into account that Aristotle did not use the term in question, we can assume that cataract surgery first appeared in Athens during the period between Aristotle’s treatise “On the Generation of Animals”—i.e., most likely the last period of his life (335–322 B.C.E.)—and the years around 250–208 B.C.E. The year 250 B.C.E. marks the possible beginning of Chrysippus’s writing career.

### 4.2. From Whom Did the Greeks Learn How to Perform a Cataract Surgery?

The conquests of Alexander the Great resulted in the collapse of the hitherto prevailing political system in Greece. The political structure based on city-states fell apart. The philosophy of Stoicism was born as a reaction to that breakdown. How do you live in a world that has virtually no boundaries? The situation created similar problems to the ones we experience today in connection with globalisation. The world had suddenly become limitless, and so it was necessary to find new measuring tools. Greek cities had become a part of a multinational state that was the Macedonian Empire. What followed was a transfer of foreign cultures and knowledge on an unprecedented scale. This is clearly visible when we look at the members of the Stoic school of philosophy. Zeno was born in Citium, Cyprus. His successor, Cleanthes, was born in Assos, today’s western Turkey. Chrysippus, the successor of Cleanthes, was born in Soli, Cicilia, today’s southern Turkey.

From where then? We propose three hypotheses. First of all, India. According to some sources, cataract surgeries were performed in India in the 6th century B.C.E. However, although Alexander the Great did reach India during his conquests, the distance between India and Greece is long enough to doubt whether Alexander came across cataract surgery specialists on his journey to the Indian subcontinent. The second hypothesis points to Egypt. It is probable that in Egypt cataract surgeries were performed before Chrysippus [23,24]. However, direct evidence is missing [25]. The third hypothesis is that the procedure was invented in Greece. At the time, contacts between Egypt and Greece were strong, especially between Athens and Alexandria. We know that Chrysippus valued Praksagoras and considered him an ally in locating the soul in the heart. We cannot exclude that Chrysippus also knew Herophilos, who believed the soul was located in the brain. However, we can go back in time even a bit further to the times of Herodotus, who said that Egyptian physicians, unlike Greek physicians, were divided into medical specialties: “The practice of medicine is so divided among them, that each physician is a healer of one disease and no more. All the country is full of physicians, some of the eye, some of the teeth, some of what pertains to the belly, and some of the hidden diseases” [26,27,28]. Furthermore, Greek medicine was non-invasive. The Greek word pharmakon(φάρμακον) means both medication and poison. According to Aristotle, a physician is someone who takes care of the patient’s health, not simply treats the disease, focusing his attention not on removing the disease but rather on restoring one’s health. That is why Greek medicine puts such an emphasis on good diet and healthy lifestyle, so to speak. Medications are treated as a transgression and should only be used when health cannot be restored with the organism’s own powers. Herodotus tells us a story that shows the difference between the Egyptian and Greek approach towards medicine. Greek medicine was gentle, while Egyptian medicine was more direct and invasive [27,28]. Of course, when a Greek author compares Greeks and non-Greeks, one cannot expect him to be impartial in his opinions. However, in this particular case it seems that Herodotus quite accurately captured the specific character of Greek medicine. For his contemporaries, a disease was a manifestation of the natural voice of the organism, while taking medications could be interpreted as ignoring or even strangling that voice. If we think about with how much resistance the ancient Greeks approached medications, we can only imagine how much they must have despised surgeries and other medical procedures. That kind of approach was also promoted by Galen, for whom surgery was the last resort [26]. We know that Greeks did not approve of autopsies. They were performed, at least for some time, in Alexandria. What is more, according to Celsus, both Herophilos and Erasistratus allegedly performed autopsies on living (!) people and animals [29]. Furthermore, we need to remember that an eye is a very specific organ, as it is partly located outside of the body. According to Calcidius [30,31], Alcmaeon did perform an eye autopsy (5th century B.C.E.) What is more, some scholars interpret the term exssectio, used by Calcidius when describing the autopsy performed by Alcmaeon, more broadly as the autopsy of the human body (usually the term dissectio was used) [32]. The story about the goat and a cataractis well known. Claudius Aelianus (c. 175–235) writes: ”The Goat, it seems, is in fact skilful at curing that mist of the eyes which doctors call ‘cataract’(hypochysin, ὑπόχυσιν), and it is even said that men have learnt this cure from the Goat. This method is as follows. When the Goat perceives that its sight has become clouded it goes to a bramble and appllies its eye to a thorn. The thorn pricks (ekentese, ἐκέντησε) it and the fluid is discharged, but the pupil remains unharmed and the Goat regains its sight without any need of man’s skill and manipulation.” [33] All in all, it seems that the pro-surgery Egyptian medicine, combined with the scientific frenzy of the Alexandrian school, would be the most probable answer to our question on the origins of cataract surgery. However, we need to emphasise that we cannot know that for sure.

We have assumed that Chrysippus first came across cataract surgery in Athens. However, there were many other countries between Egypt and Greece at that time and even more between Greece and India. If we are looking for the origins of the cataract surgery in India and Egypt, maybe we should also look for them in those countries that lie between the two. Chrysippus, as has already been mentioned, came from Soli or Cicilia, or from Tarsus, according to other sources. What is more, Zeno, the founder of the Stoic school of philosophy, was a Phoenician from Citium, a Greek colony on Cyprus. His father was a merchant. Although the only information that we have is that Zeno’s father travelled to Athens, in those times good command of geography was a prerequisite for being a merchant [9]. That is why we cannot rule out the possibility that one of them might have come across the surgery in Asia Minor. However, if Chrysippus had heard about cataract surgery second-hand or if he had learned about it during one of his trips abroad, the argument in which the cataract plays the major part would have to be weaker. After all, in his text, Chrysippus is not writing about the wonders of faraway lands but is looking for examples that would validate his arguments. We learn from Diogenes Laertius “that most people thought, if the gods took to dialectic, they would adopt no other system than that of Chrysippus” [11]. Thus, if Chrysippus’s arguments were really worthy of gods, cataract surgery would have to be known to Athenians. Therefore, if medical knowledge travelled to Athens, whether it was from India or Egypt, it must have “visited” other places along the way. Chrysippus’s testimony shows that in his time, Athenians already knew about cataract surgery. 

We owe Celsus the first medical description of cataract extraction in the chapter 7th of his “De Medicine” [29]. There is also a very old misunderstanding related to this treatise often repeated in the literature [34,35,36,37] that he delivered in the introduction to the chapter 7th the information that the first cataract surgery was conducted by Philoxenus in Egypt in the period between the 4th and 3rd centuries B.C.E. We correct this—there is no such information in his treatise. Celsus mentioned Philoxenus only once in the following part: “This branch [medicine which cures by hand], although very ancient, was more practised by Hippocrates, the father of all medical art, than by his forerunners. Later it was separated from the rest of medicine, and began to have its own professors; in Egypt it grew especially by the influence of Philoxenus, who wrote a careful and comprehensive work on it inseveral volumes” [29]. This fragment is not related to eye surgery but generally to surgery, and in the other part where cataract surgery is described, Celsus did not mention Philoxenus. Interstingly, we were able to find that this misundersting originates at least from the early 19th century. In 1818, this was proposed by Percy [38] and later corrected in 1847 [39]. Galen mentioned Philoxenus and named him Claudius Philoxenus [40], which suggested that he lived during Roman times, much later than famous Alexandrian school of medicine from the 3rd century B.C.E.It is very probable that both referred to a Greco-Egyptian physician who lived in Alexandria in the 1st century CE.

### 4.3. The Philosophical Context of Chrysippus’s Text

It is hard to establish whether Chrysippus wrote about cataract in any other of his works. Unfortunately, the catalogue of his texts provided to us by Diogenes Laërtius ends on his papers on ethics. We learn that Chrysippus’s main area of study was logic, to which the philosopher dedicated 311 of his works [11]. The surviving part of the catalogue shows that his second main field of study was ethics, which is characteristic of all Stoics. The fragment that is the object of this article was probably a part of a dialectic treatise. Arnim, the author of a collection of fragments written by Stoic philosophers, located the fragment in the sub-section “On the Contraries” (Περὶ ἐναντίων), just like several [41] earlier fragments from pages 98–102 of the Simplicius’s commentary. 

The arguments used by both Chrysippus and Aristotle show that vision, though it was not the object of their studies, was used as a convenient, albeit dangerous, metaphor. One could go as far as to say that it was one of their beloved analogies. Aristotle starts his “Metaphysics”with these words: “All men naturally desire knowledge. An indication of this is our esteem for the senses; for apart from their use we esteem them for their own sake, and most of all the sense of sight” [42]. We usually forget that the major concept of Plato’s philosophy, idea/eidos, is closely related to the verb eidon—“I saw”—and to the Indo-European root vid*, or visible—e.g., as in the Polish widzę (“I see”). On one hand, such an approach, already visible from the heights of the Olympic pantheon, must have been rich in metaphors. On the other hand, when looking for medical knowledge in philosophical texts, we need to remember that for the Greek philosophers, vision, just as much as a lack thereof, is primarily treated as a specific and divine moment in which humans become god-like. However, it is also a moment in which people make mistakes, and no other sense was believed by the Greeks to be so dangerous.

## 5. Conclusions

First of all, the analysis of the text by Simplicius shows that cataract surgery was known to Chrysippus and that it was performed in Athens in the 3rd century B.C.E. We were not able to establish who taught the Greeks how to remove a cataract. The oldest reference to cataract surgery in the available literature is not found in a medical text but in a Greek philosophical one. Interestingly, the terminology used by Chrysippus was later applied by specialists in the field, including Celsus in the 1st century C.E. [29], and served as a basis for the Latin terminology.

## Figures and Tables

**Figure 1 medicines-07-00034-f001:**
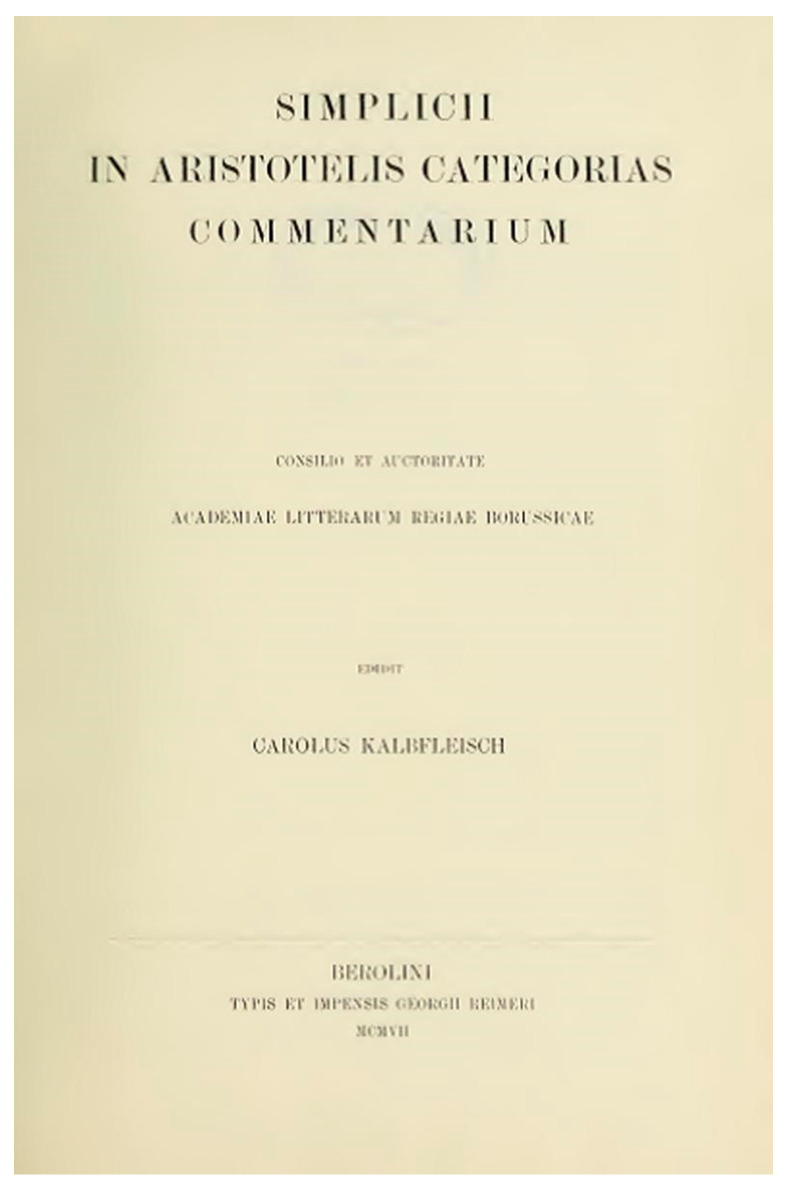
The title page of Simplicius’s commentary. 1907 edition, 8th Volume of the Greek Commentaries to Aristotle.

**Figure 2 medicines-07-00034-f002:**
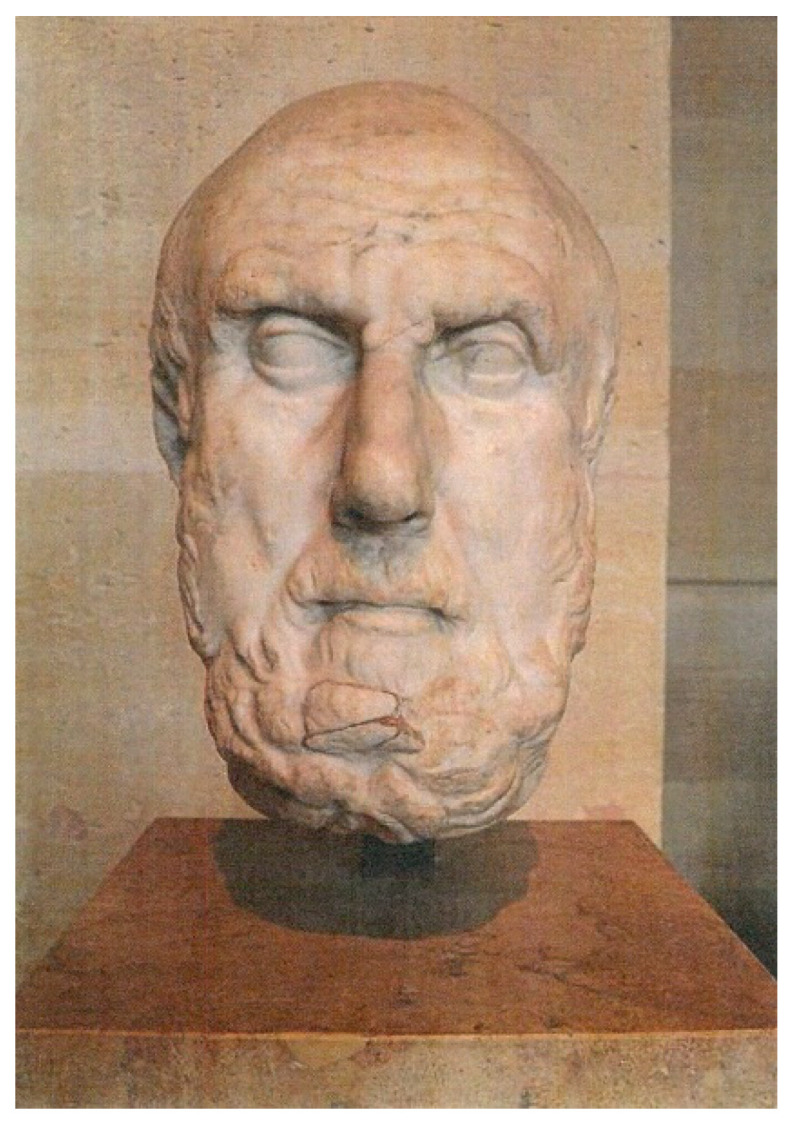
Portrait of Stoic philosopher Chrysippus [Source: https://commons.wikimedia.org/wiki/File:Portrait_of_the_Stoic_philosopher_Chrysippus,_2nd_century_AD,_Louvre_Museum_(7462805964).jpg].

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
