# Peer review of "Chrysippus and the First Known Description of Cataract Surgery"

_medicines, 2020, doi:10.3390/medicines7060034_

Round 1
Reviewer 1 Report
This manuscript is more philosophical than ophthalmological. The ocular aspects are fine except for the statement that the eye is partially outside the body, which is not clear.
The aspects of Aristotle and Stoic philosophy are discussed by Prof Eric Brown, a member of the department of philosophy at Washington University in St Louis, who is an expert in this field. His questions should be addressed.
Author Response
We thank the reviewer for the valuable comments. We added a comment regarding the role of Stoicism referenced to prof. Eric Brown.
Reviewer 2 Report
The present work is of great historical value and a meticulous study of chysippus and the first cataract surgery. Although the philosophical discussion is not always easy for an ophthalmic surgeon, the historical evaluation in this work is important and significant for understanding a long history of cataract surgery. From my point of view there are no further comments.
Author Response
We would like to thank the reviewer for his opinion of our work.